# Association of Guideline-Based Medical Therapy with Malignant Arrhythmias and Mortality among Heart Failure Patients Implanted with Cardioverter Defibrillator (ICD) or Cardiac Resynchronization-Defibrillator Device (CRTD)

**DOI:** 10.3390/jcm10081753

**Published:** 2021-04-17

**Authors:** Tal Hasin, Ilia Davarashvili, Yoav Michowitz, Rivka Farkash, Haya Presman, Michael Glikson, Moshe Rav-Acha

**Affiliations:** 1Jesselson Integrated Heart Center, Shaare Zedek Medical Center, Hebrew University, Jerusalem 9103102, Israel; hasint@szmc.org.il (T.H.); idavarashvili@yahoo.com (I.D.); yoavm@szmc.org.il (Y.M.); rivka_f@szmc.org.il (R.F.); mglikson@szmc.org.il (M.G.); 2Faculty of Medicine, Hebrew University, Jerusalem 9103102, Israel; Haya.presman@mail.huji.ac.il

**Keywords:** heart failure, medical treatment, defibrillator, arrhythmia

## Abstract

Aim: Evaluate prevalence of heart failure (HF) medications and their association with ventricular arrhythmia (VA) and survival among patients implanted with primary prevention implantable cardiac defibrillator (ICD)/cardiac resynchronization therapy + defibrillator (CRTD) devices. Methods: Association of treatment and dose (% guideline recommended target) of beta-adrenergic receptor antagonist (BB), angiotensin-antagonists (AngA), and mineralocorticoid-antagonists (MRA), after ICD/CRTD implant with VA and mortality was retrospectively analyzed. Results: Study included 186 HF patients; 42.5% and 57.5% implanted with ICD and CRTD, respectively. During 3.8 (2.1;6.7) years; 52 (28%) had VA and 77 (41.4%) died. Treatment (% of patients) included: BB (83%), AngA (87%), and MRA (59%). Median doses were 25(12.5;50)% of target for all medications. BB treatment >25% target dose was associated with reduced VA incidence. In the multivariable model including age, gender, diabetes, heart rate, and medication doses, increased BB dose was associated with reduced VA (hazard ratio (HR) 0.443 95% CI 0.222–0.885; *p* = 0.021). In the multivariable model for overall mortality including age, gender, renal disease, VA, and medical treatment, VA was associated with increased mortality (HR 2.672; 95% CI 1.429–4.999; *p* = 0.002) and AngA treatment was associated with reduced mortality (HR 0.515; 95% CI 0.285–0.929; *p* = 0.028). Conclusions: In this cohort of real-life HF patients discharged after ICD/CRTD implant, prevalence of guideline-based HF medications was high, albeit with low doses. Higher BB dose was associated with reduced VA, while AngA was associated with improved survival.

## 1. Introduction

Adherence to heart failure (HF) recommended medical treatment guidelines was shown to reduce HF symptoms, HF hospitalizations, and all-cause mortality in multiple publications [1,2,3,4,5,6,7,8,9,10]. Although the impact of such treatment on reduced ventricular arrhythmia (VA) was suggested [1,2,8,11,12,13], this was not evaluated as the primary outcome in randomized trials but rather as a secondary outcome [1,2,8] or in the context of meta-analysis [11,12,13,14]. Moreover, most previous studies did not investigate the dosing relation of HF medications with VA or sudden cardiac death (SCD). Circumstantial evidence suggests that combination HF therapy reduces SCD rate and might mitigate the added survival benefit of an implantable cardiac defibrillator (ICD) device among HF patients in general and specifically among non-ischemic dilated cardiomyopathy (DCM) patients, in whom the evidence for survival benefit with an ICD is weaker [15,16]. A meta-analysis of pivotal HF trials has shown a continuous decline of SCD incidence as the trials became more recent. This observation was attributed to the increased utilization of HF guideline-based medications in recent trials compared with older ones [15]. Moreover, among DCM patients in the DANISH trial [16] there was no significant mortality difference between patients treated with optimal medical management including cardiac resynchronization therapy (CRT) as appropriate and those treated similarly with additional ICD. Again, suggesting that current guideline-based medical therapy may obviate the need of an ICD in selected patients. This finding was reinforced in a recent meta-analysis of randomized trials evaluating the survival benefit of ICD in DCM patients, revealing loss of the survival benefit in trials where >50% of patients were taking a combination of beta-adrenergic receptor antagonist (BB), angiotensin antagonist (AngA) including Angiotensin Converting Enzyme Inhibitors and Angioentsin Receptor Blockers, and mineralocorticoid receptor antagonist (MRA) [14]. Nevertheless, the direct (non-circumstantial) impact of HF disease modifying medications and doses on VA as a primary outcome needs further establishment. Noteworthy, in contrast with the above-mentioned HF trials, large registries of HF patients have shown relatively low percent of patients treated with optimal HF medical therapy [17,18,19,20].

The aim of the current study was to evaluate the prevalence of HF medical therapy and its direct association with VA incidence and overall mortality, with relation to medication dosage, among contemporary primary prevention ICD/cardiac resynchronization therapy + defibrillator (CRTD) recipients.

## 2. Methods

Retrospective single center analysis of HF patients implanted with primary prevention ICD/CRTD to evaluate the association between HF modifying drugs, with emphasis on their doses, to VA and total mortality.

### 2.1. Study Patients

HF patients hospitalized at Shaare Zedek Medical Center between the years 2007–2017 for de novo ICD or CRTD implant and who were followed at our hospital’s device clinic were included.

Inclusion criteria were therefore:Primary prevention implant of an ICD or CRTD;At least 4 device clinic visits during the study follow up period.

Exclusion criteria were:Device upgrade during the study follow-up period;Implant at another center (incomplete device interrogation data);Previous sustained VA or cardiopulmonary resuscitation.

Eligible patients’ data were retrospectively reviewed by a senior cardiologist that confirmed their indication for primary prevention ICD/CRTD according to current guidelines [21,22,23], verifying symptomatic HF, low (≤35%) systolic left ventricular ejection fraction (EF), and the absence of exclusion criteria. Medical treatment was determined based on medical prescriptions in the discharge letter of the index hospitalization (hospitalization in which ICD/CRTD was implanted). Guideline-recommended disease modifying HF medications were grouped according to mechanism of action as beta-adrenergic receptor antagonist (BB), angiotensin antagonists (AngA) including angiotensin receptor blockers (ARB) or angiotensin conversion enzyme inhibitors (ACE-I), and mineralocorticoid receptor antagonist (MRA). All anti-arrhythmic drugs (AAD) used were documented as well. The proportion of each HF medication dose to the guideline recommended target dose [23,24] was calculated and reported as % target dose. As the median dose for all 3 medication groups analyzed in our study was 25% of guideline-recommended target dose, we used the median dose cutoff to examine the effect of medications’ doses on study outcomes. Renal dysfunction was defined by Glomerular Filtration Rate (GFR) < 50 mL/min.

### 2.2. Outcomes

Outcomes included VA and all-cause mortality. Follow up for outcomes was initiated from the index hospitalization, when ICD/CRTD was implanted, until mortality or last documented visit to HF or device clinic. VA was defined as any VA episode for which an appropriate anti-tachycardia pacing (ATP) or shock therapy was delivered by the ICR/CRTD device, as detected during device clinic follow-up. Device clinics were routinely scheduled 1,3, and every 6 months after device implant, although many patients did not follow these scheduled visits and their actual follow up visits varied widely in time. During clinic visits, all VA episodes, necessitating device treatment (ATP, shock, or both) were documented. When multiple VAs occurred, the first one was considered for study outcomes. Devices were programmed in a “primary prevention” mode (similar in all device companies), in accordance with the updated expert consensus on optimal ICD programming [25], consisting of the following detection zones and therapies: VF therapy zone > 200–220 bpm for 24–30 beats, treated via ATP during charge and thereafter device shocks; VT_2_ therapy zone > 185 bpm for 30 beats or 12 s duration (BSC devices), treated via ≥1 ATP burst and thereafter device shocks; and a VT_1_ monitor zone, which varied between a lower detection rate of 140 bpm in some patients to lower detection rate of 160 bpm in others. Mortality was determined from the Israeli Ministry of the Interior records. The study was approved by the local institutional review board.

### 2.3. Statistical Analysis

Categorical data are represented as proportions, continuous data as mean ± SD for normally distributed variables or median and interquartile range for non-normal distribution. Comparisons were made using chi-square test, Fisher’s exact test, unpaired student T-test and Mann–Whitney test. Multivariable Cox proportional hazard models were used to identify independent characteristics and medical treatment associated with VA or mortality. To assess the impact of VA on overall mortality, a Cox model with time to first VA as a time dependent covariate was used. Unadjusted and adjusted hazard ratios (HRs) with 95% confidence intervals (CIs) were displayed. All tests were two-sided, *p*-values < 0.05 were considered statistically significant. Analyses were carried out using IBM SPSS Statistics for Windows, Version 25.0. Armonk, NY, USA.

## 3. Results

There were 186 patients implanted with an ICD/CRTD between the years 2007–2017 that matched the study’s inclusion criteria (Figure 1). Their mean age was 66.4 ± 12 years, 15.1% were female. ICD was implanted in 79 (42.5%) and a CRTD in 107 (57.5%). Median (IQR) follow-up time was 3.8 (2.1–6.7) years. Patient characteristics are shown in Table 1. There were 52 (28%) patients with VA, including VT in 31/52 patients (59.5%), VF in 6/52 patients (11.5%), or both in 15/52 patients (29%). These VA cases were treated successfully by anti-tachycardia pacing (ATP) in 22 (42.4%) patients and by device shock in 30 patients (57.6%). There were 77 (41.4%) deaths during the study follow-up period. The prevalence of HF medication treatment at index hospitalization discharge was: 155/186 (83.3%) BB, 162/186 (87.1%) AngA, and 110/186 (59.1%) MRA. AADs were prescribed in 81/186 (43.5%) patients, including 52/81(64.2%) patients on amiodarone, 29/81(35%) on digoxin and 6/81(7.4%) on sotalol (few with combination of AADs). Doses (% target) of HF medications were: 32 ± 25% for BB, 38.2 ± 30% for AngA and 31 ± 30% for MRA. The median dose (% target dose) for all three guideline-based medication groups included in our study was 25% (Table 2). Few patients were prescribed with >50% of target dose: 18/155 (11.6%), 34/162 (21%), and 16/110 (14.5%) of patients taking >50% target dose of BB, AngA, and MRA, respectively (Table 2).

Only 18/186 (9.7%) of study patients were followed regularly in the hospital’s HF clinic by an HF specialist (most patients were followed regularly by their general cardiologists and came to our hospital only for device clinic interrogations). There were more patients treated by BB among the group followed in HF clinic (100% vs. 81.5%, *p* = 0.046) and their dose (% target dose) was higher (61.1% vs. 33.9%, *p* = 0.023). There was a non-significant trend for higher prevalence of AngA (88.9% vs. 86.9%, *p* = 0.81) and MRA (72.2% vs. 57.7%, *p* = 0.23) among those followed at the HF clinic as well.

### 3.1. Association of HF Medical Treatment with VA

Comparing patients with documented VA to those without VA, revealed similar baseline characteristics, except for lower prevalence of diabetes mellitus (DM) and longer follow-up among the VA group (Table 1). Crude medication prescription was not associated with VA, nor was the number of guideline-based medications (2.3 ± 0.83 for VA vs. 2.25 ± 0.7 without VA; *p* = 0.8). Patients taking all three guideline-recommended medication groups did not have less VA (*p* = 0.33). The patients with VA were treated with significantly lower doses of BB compared to those without VA (23.9 ± 19% versus 35.5 ± 27% target dose; *p* = 0.012). There was no significant difference in AngA or MRA doses between the VA and no-VA groups (Table 1).

The VA incidence was significantly reduced among patients treated by >25% target dose of BB as compared to those treated with ≤25% target dose (17.6% vs. 33.9%, *p* = 0.017). This was not observed in patients taking >25% AngA (30% vs. 26%, *p* = 0.55) or MRA (29.5% vs. 26.5%, *p* = 0.64) compared to those treated by ≤25% target dose of these medications. Kapkan–Meier (KM) analysis for survival without VA according to each medication group dose, supported reduced VA among patients receiving >median-dose of BB (Figure 2).

Univariate parameters found to be significantly associated with VA incidence were: heart rate at admission (HR 1.02; 95% CI 1.00–1.04; *p* = 0.02), DM (HR 0.42; 95% CI 0.23–0.78; *p* = 0.006), and BB >25% target dose (HR 0.51; 95% CI 0.27–0.98; *p* = 0.04). In the Cox multivariable model for VA including age, gender, DM, medication dosage (>25% target dose), and heart rate, both BB dose > median dose (HR 0.443, 95% CI 0.222–1.022; *p* = 0.021) and DM (HR 0.454, 96% CI 0.237–0.868; *p* = 0.017) were significantly and independently associated with lower incidence of VA; while increased heart rate was significantly associated with increased incidence of VA (HR 1.03, 95% CI 1.009–1.049; *p* = 0.004) (Table 3).

### 3.2. HF Medication Treatment and Overall Survival

Potential predictors of mortality are presented in Table 4. Older age at device implant, renal dysfunction, documented atrial fibrillation, CRTD implant (rather than ICD), and VA episodes during follow-up were associated with increased overall mortality. Analysis of HF medications showed that combined treatment with all three HF medication groups (*p* = 0.0047) and treatment with AngA per se (*p* = 0.028), regardless of dose, were significantly associated with reduced mortality (Table 4).

In a univariate analysis the following parameters were significantly associated with overall mortality: age (HR 1.06; 95% CI 1.04–1.09; *p* = 0.0001), renal dysfunction (HR 1.63; 95% CI 1.03–2.56; *p* = 0.037), CRTD (versus ICD) (HR 1.67; 95% CI 1.03–2.71; *p* = 0.036), VA during follow-up (HR 2.76; 95% CI 1.474–4.967, *p* = 0.001), and AngA treatment (HR 0.55; 95% CI 0.31–0.97; *p* = 0.039).

In Cox multivariable survival analysis including patients’ ages, genders, renal function, HF medication treatments, and VA occurrence during follow-up, AngA treatment (but not BB or MRA) was significantly associated with reduced mortality (HR 0.515; 95% CI 0.285–0.929; *p* = 0.028); while age (HR 1.06; 95% CI 1.038–1.093; *p* = 0.0001); renal disease (HR 1.728; 95% CI 1.070–2.792; *p* = 0.025); and VA during follow-up (HR 2.672; 95% CI 1.429–4.999; *p* = 0.002) were significantly associated with increased mortality (Table 5).

Kaplan–Meier overall survival analysis according to HF medication groups showed reduced mortality among patients treated by AngA (*p* = 0.036) without significant impact of BB or MRA treatment (Figure 3). Interestingly, Kaplan–Meier overall survival curves for the combination of all three HF medication groups diverged for improvement with combined treatment after four years (curve not shown). Kaplan–Meier overall survival curves by incident VA (as a competing event) revealed increased mortality in patients with VA (Figure 4).

## 4. Discussion

This study, including 186 HF patients implanted with a primary prevention ICD or CRTD and meticulously followed by the device clinic, evaluated the impact of guideline-based HF medications on incident VA and total mortality. During the median follow-up period of 3.8 years, 28% of the patients had VA and 41.4% died. On the whole, although most of the patients were prescribed with the appropriate HF medications (>80% for BB and AngA and 60% for MRA), the doses were low. The median dose of HF medications in the current study was 25% of target dose for all three medication groups with less than 20% of patients treated by >50% target dose. Many patients (43.5%) were treated by AAD, mostly amiodarone and digoxin, due to Atrial Fibrillation AF, which is prevalent in advanced HF patients [16,17]. We found that treatment with lower doses of BBs and increased heart rates were both significantly and independently associated with increased VA, while DM was associated with reduced VA incidence. We also found that treatment with AngA was significantly associated with reduced overall mortality, while VA and renal dysfunction were associated with increased mortality.

The incidence of VA in the current study is comparable to previously published studies. In the SCD-HeFT primary prevention trial which had a similar follow-up period, the incidence of appropriate ICD shocks was 21.5% [26]. The estimated annual incidence of VA in our study of 7.4% is similar to the 7.2% annual appropriate shock incidence in the DEFINITE primary prevention trial [27]. Notably, the patients’ devices in the current study were routinely programmed via prolonged VA detection periods to enable spontaneous termination of short VAs, as well as device intervention for relatively fast VAs. Thus, only long and fast VAs were included in the current study. Importantly, these clinically relevant VAs do not equal sudden cardiac death, as they might still end spontaneously [28,29,30]. Nevertheless, these VAs do have a significant impact on overall mortality, as was shown in the current study and as supported by several prior studies establishing the benefit of ICD implant [26,27,31,32,33,34].

In the current study the dosage of BB, rather than their mere use, was associated with VA reduction. The importance of aiming for target doses to decrease mortality and HF hospitalization was shown in multiple studies, revealing either increased deaths, HF hospitalizations, or both, among HF patients treated with <50% target dose of BB [7,20,24,35]. The importance of HF medication dosage was further emphasized in the DANISH trial where optimal medical therapy, with medication prevalence of >90% for BB and AA and 60% for MRA (similar to current study) and doses that were increased to target whenever possible, were suggested to obviate the survival benefit of an ICD [16]. In contrast, there is a paucity of studies examining the direct relation of BB dosage to VA, as a primary outcome [36,37]. Moreover, only few previous studies examined the impact of BB dose on VA among a specific subpopulation of primary prevention ICD/CRTD recipients. Among these, a retrospective analysis for MADIT II ICD recipients, revealed reduced appropriate ICD therapies among patients treated by BB in the higher quartile dosage [38]. The Danish nationwide cohort study including DANISH trial patients showed a significant relation between increased BB dose and reduced VA, HF hospitalizations, and death among primary prevention ICD recipients [39]. Similarly, up titration of BB dose in HF patients undergoing primary prevention CRTD implant, was recently shown to reduce device appropriate therapy for VA [40]. On the whole, our study re-emphasizes the importance of HF medication dosage in general and BB dose specifically, for reducing VA in advanced HF patients implanted with an ICD or CRTD.

Potential mechanisms for antiarrhythmic effects of BB include their anti-sympathetic effect resulting in reduced heart rates and increased heart rate variability, direct anti-arrhythmic effect, reducing intra-cellular Ca within cardiac cells, improving cardiac function, reducing cardiac ischemia, and more [41,42,43]. In the current study, reduced BB doses were associated with increased VA even when adjusting for heart rate. Therefore, BBs may have an anti-arrhythmic effect beyond decreasing heart rate per se. This result is in line with previous trials [44,45,46], which show that although increased heart rates are associated with worst outcomes in HF patients, increasing BB dose regardless of baseline heart rates is associated with an improved combined outcome of all-cause mortality and HF hospitalizations. Accordingly, we as others [39,44,45,46], suggest that BB dose up-titration, regardless of baseline heart rate, should be considered for VA prevention (as long as symptomatic or excessive bradycardia is absent).

Low dosing of all HF recommended medications was one of the main findings in the current study. Low dosing was noticed in multiple HF studies and registries [17,18,19,20,24], acknowledging that this is a universal problem. For example, in the CHAMP-HF registry including 3500 HF patients with reduced EF from 150 medical centers, less than 25% of patients received target doses of any HF medication and only 1% received target doses of all three HF family medications [17,18]. Similarly, only a minority of patients in the Asian [19] and pan-European [24] registries received target doses of any HF medication. Low dose HF medications could result from inadequate medical surveillance, non-referral to specialized HF clinics, or otherwise impacted by various “obstacles” such as low blood pressure or heart rate, comorbidities, and medication-related side effects preventing one from achieving target doses. In the current study, patients with and without VA had similar comorbidities, with Hypertention (HTN) in most patients and heart rates between 70 and 80 bpm in both groups. Hence, although we do not have the specific reason for low medication dosage on our study patients, we suggest that the lower BB dosage among patients with VA is not related to sicker patients who cannot tolerate increased BB doses but rather suboptimal medical surveillance. This is in line with the limited number of study patients who were followed in the HF clinic, where significantly more patients were treated by BB with increased doses and trends for increased prevalence of AngA and MRA medications as well. Importantly, as we do not have the clear reason for low medication dosage in each patient, we could not rule out low dosage secondary to advanced HF stage. Nevertheless, we do suggest from the above circumstantial data, that this was not the case.

The importance of HF consultation prior to device implant should be emphasized as in reality many ICD/CRTD candidates are referred to cardiac electrophysiology (EP) clinic by their general cardiologists or GPs for device implantation without HF consultation and with inadequate HF medical treatment. Thus, we suggest that all HF patients, and especially those referred for device implant, undergo HF specialist consultation, aiming to achieve HF medication target doses. Importantly, this approach is strongly supported by both EP and HF guidelines advocating ICD or CRTD implant only after confirmation of optimal HF medical treatment [21,22,23].

## 5. Strengths and Limitations

Our study has several limitations including: (a) its retrospective nature; (b) single center data with low patient volume, exposed to type 1 statistical errors; (c) the overall low doses of guideline-based medications, resulting in possible underestimation of medication effect; (d) discharge prescriptions may not equal true medical treatment over time. Aalthough most patients remain treated with their discharge recommendations, analysis of medication compliance and consistency could not be performed due to widely variable clinic visit timings, precluding homogenous time analysis of medications for all patients; (e) most study patients were included prior to 2017 and thus were not treated with angiotensin receptor/neprilysin inhibitors (ARNI). Thus, our study did not evaluate the impact of ARNI, which is a pivotal HF medication in recent years, nor were they treated by SGLT2-inihibitors, which were recently shown to be HF modifying drugs as well; (f) only VA necessitating ICD/CRTD interventions were evaluated. Apparently, some VA had occurred in a monitor zone not necessitating any device intervention and thus were not included in our outcomes; and (g) data on cause of death is missing. The study also has several strengths including the meticulous retrieval of “serious” VA events and the in-depth manual evaluation of discharge medication dose analyzed as the proportion of guideline recommendations.

## 6. Conclusions

In this single center retrospective cohort of HF patients implanted with an ICD/CRTD for primary prevention, we found a relatively high prevalence of HF guideline-recommended medication treatment, albeit with low doses. Reduced BB doses were associated with increased Vas, which in turn are associated with increased mortality, while treatment with AngA was associated with reduced overall mortality. Specialized HF consultation is therefore advocated for these patients referred for primary prevention ICD/CRTD to improve their medical treatment and outcomes.

## Figures and Tables

**Figure 1 jcm-10-01753-f001:**
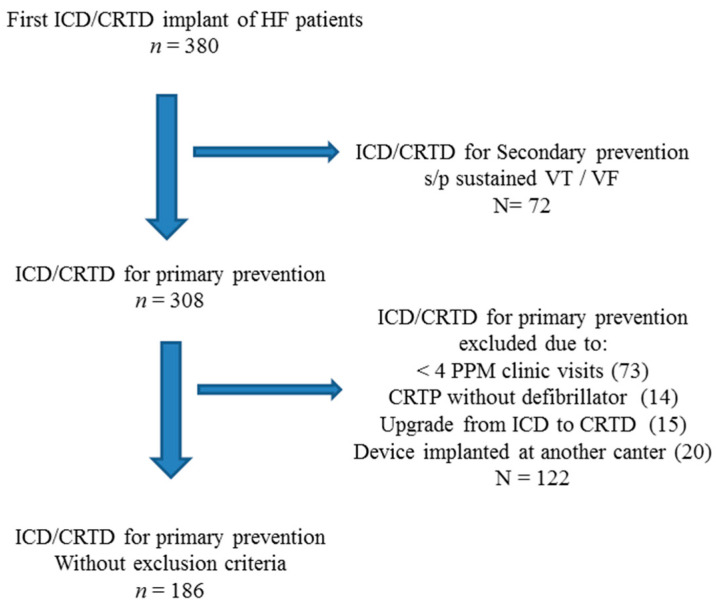
Study subjects depicting inclusion and exclusion criteria. ICD= Implantable Cardioverter Defibrillator; CRTP= Cardiac Resynchronization Therapy; CRTD= Cardiac Resynchronization Therapy with Defibrillator; PPM= permanent pacemaker.

**Figure 2 jcm-10-01753-f002:**
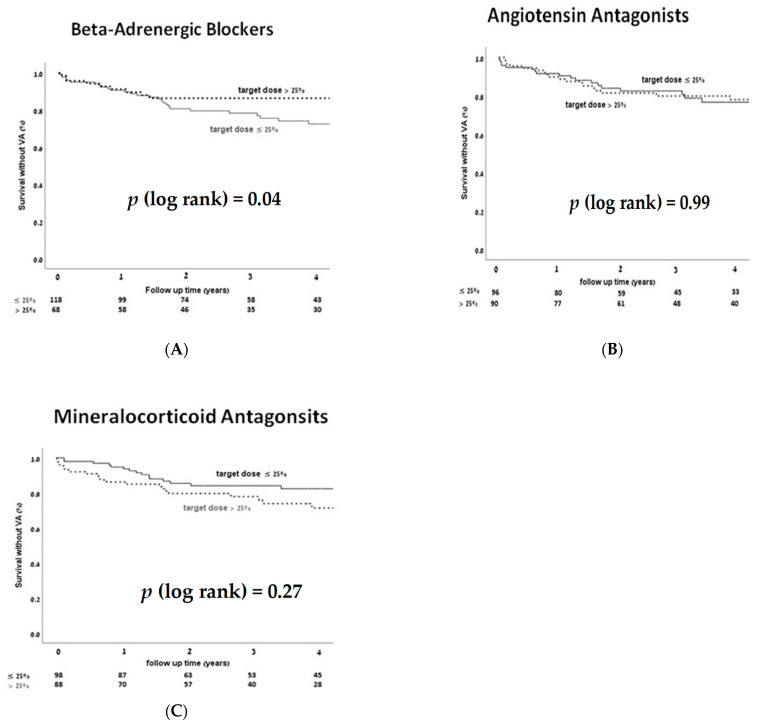
Survival without ventricular arrhythmia (VA) Kapkan–Meier (KM) curves according to heart failure medications dose (> or ≤ median dose), for each HF medication group including BB (**A**), AngA (**B**), and MRA (**C**). There was significantly less VA among patients taking > median dose of BB, with no significant impact of AngA or MRA medication dosages on VA occurrence. *p* < 0.05 is considered significant.

**Figure 3 jcm-10-01753-f003:**
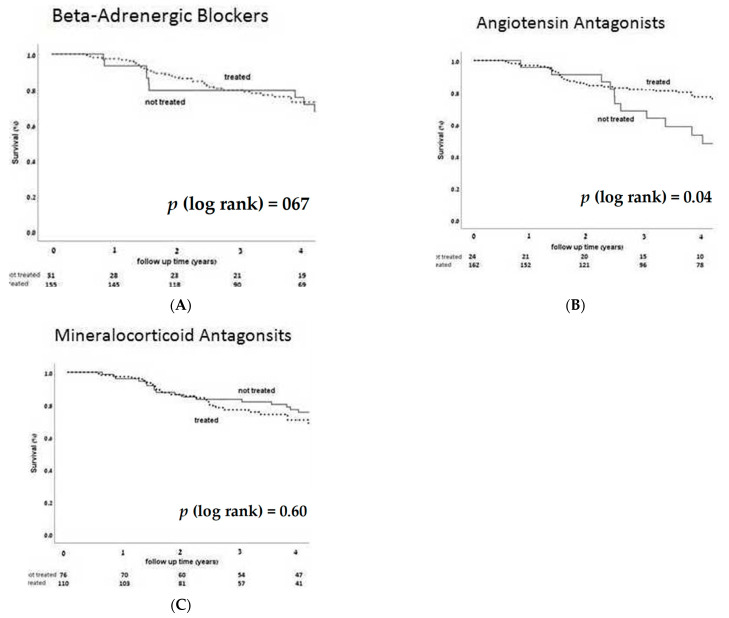
Overall survival KM curves according to heart failure medication treatment groups (regardless of dose), including BB (**A**), AngA (**B**), and MRA (**C**), revealing reduced mortality in patients treated with angiotensin antagonists. *p* < 0.05 is considered significant.

**Figure 4 jcm-10-01753-f004:**
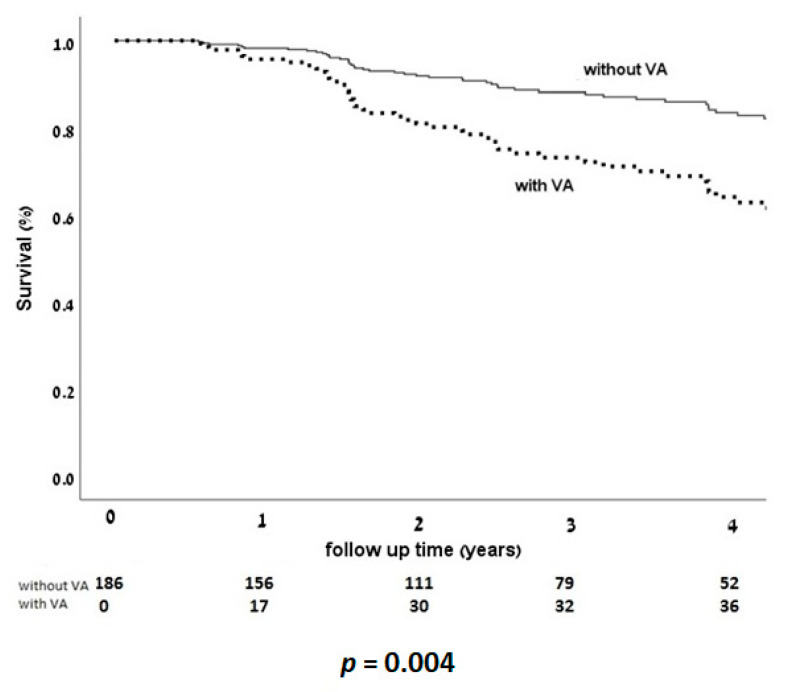
Survival KM curves according to presence or absence of VA, as a time dependent covariate. Occurrence of VA had a significant impact, increasing overall mortality. *p* < 0.05 is considered significant.

**Table 1 jcm-10-01753-t001:** Patient characteristics and comparison of patients with and without VA.

Parameter	Total (*n* = 186)	VA (*n* = 52)	No VA (*n* = 134)	*p*
Age	66.4 ± 12	66.8 ± 11.8	66.7 ± 11.7	0.7
Gender (male)	158 (84.9%)	48 (92.3%)	110 (82.1%)	0.08
Heart Rate (admission)	72.5 ± 14	75.7 ± 16	71.4 ± 12	0.068
ICD	79 (42.5%)	26 (50%)	53 (39.6%)	0.22
CRTD	107 (57.5%)	26 (50%)	81 (60.4%)	0.2
Ischemic CM	115 (61.8%)	29 (55.7%)	86 (64.1%)	0.25
HTN	130 (69.8%)	35 (67.3%)	95 (70.8%)	0.7
DM	79 (42.4%)	14 (26.9%)	65 (48.5%)	0.008
Renal dysfunction	66 (35.4%)	17 (32.7%)	49 (36.5%)	0.7
Atrial fibrillation	73 (39.2%)	19 (36.5%)	54 (40.3%)	0.74
Number of guideline-based medications	2.3 ± 0.75	2.3 ± 0.83	2.25 ± 0.7	0.8
BB treatment	155 (83.3%)	40 (79.6%)	115 (85.8%)	0.14
BB dose (% target)	32 ± 25%	23.9 ± 19%	35.5 ± 27%	0.012
BB > 25% target dose	68 (36.5%)	12 (23%)	56 (41.8%)	0.084
AngA treatment	162 (87.1%)	46 (88.5%)	116 (86.6%)	0.73
AngA dose (% target)	38.2 ± 30%	41.2 ± 32%	37 ± 30%	0.38
AngA > 25% target dose	90 (48.3%)	27 (51.9%)	63 (47%)	0.943
MRA treatment	110 (59.1%)	31 (59.6%)	79 (59%)	0.93
MRA dose (% target)	31 ± 30%	32.2 ± 30.8%	30.4 ± 30%	0.74
MRA > 25% target dose	88 (47.3%)	26 (50%)	62 (46.3%)	0.932
AAD treatment	81 (43.5%)	22 (42.3%)	59 (44%)	0.83
Amiodarone treatment	52 (30%)	11 (21.2%)	41 (30.6%)	0.21
Digoxin treatment	29 (15.6%)	10 (19.2%)	19 (14.2%)	0.5
BB+AngA+MRA treatment	86 (46.2%)	21 (40.4%)	65 (48.5%)	0.33
Follow up period (median (IQR), days)	1399 (752, 2432)	1819 (930, 3140)	1326 (679, 2113)	0.005

VA-Ventricular Arrhythmia; ICD-Implantable Cardioverter Defibrillator; CRTD-Cardiac Resynchronization Therapy with Defibrillator; CM-Cardiomyopathy; HTN-Hypertension; DM-Diabetes mellitus; BB-Beta-adrenergic receptor antagonist; AngA-Angiotensin-antagonists; MRA-mineralocorticoid-antagonists; AAD-Anti Arrhythmia Drugs; IQR-Interquartile Range.

**Table 2 jcm-10-01753-t002:** Heart Failure (HF) medication groups prevalence and doses.

Medication	Prevalence *n* (%)	Median (IQR) Dose (% Target)	Dose (% Target) Average ± SD	Patients Receiving > 50% Target Dose
BB	155 (83.3%)	25 (12.5; 50)	32 ± 25%	18/155 (11.6%)
AngA	162 (87.1%)	25 (12.5; 50)	38.2 ± 30%	34/162 (21%)
MRA	110 (59.1%)	25 (0; 50)	31 ± 30%	16/110 (14.5%)

**Table 3 jcm-10-01753-t003:** Cox proportional-hazards multivariate model for ventricular arrhythmia.

Parameter	HR	95% CI	*p*
Age upon admission (years)	0.999	0.977–1.022	0.944
Gender (male)	0.388	0.138–1.092	0.073
Diabetes mellitus	0.454	0.237–0.868	0.017
Heart rate admission	1.029	1.009–1.049	0.004
BB dose > 25% target dose	0.443	0.222–0.885	0.021
AngA dose > 25% target dose	1.010	0.559–1.827	0.973
MRA dose > 25% target dose	1.407	0.783–2.528	0.254

**Table 4 jcm-10-01753-t004:** Characteristics of patients who died or survived follow-up.

Parameter	Died (*n* = 77)	Survived (*n* = 109)	*p*
Age (years)	71.1 ± 11	63.1 ± 11.6	0.0001
Male	70 (90.9%)	88 (80.7%)	0.063
ICD	25 (32.4%)	54 (49.5%)	0.024
CRTD	52 (67.5%)	55 (50.4%)	0.024
Ischemic cardiomyopathy	54 (70.1%)	61 (55.9%)	0.065
Hypertension	56 (72.7%)	74 (67.8%)	0.5
Diabetes mellitus	28 (36.3%)	51 (46.7%)	0.17
Renal dysfunction	34 (44.2%)	32 (29.3%)	0.043
Atrial fibrillation	37 (48%)	36 (33%)	0.05
VA during F/U	31 (40.2%)	21 (19.2%)	0.0028
Number of guideline-based medications	2.13 ± 0.75	2.41 ± 0.74	0.007
BB treatment	61 (79.2%)	94 (86.2%)	0.23
BB dose (% target)	29.3% ± 24%	34.3 ± 27%	0.243
BB > 25% target dose	26 (33.8%)	42 (38.5%)	0.506
AngA treatment	62 (80%)	100 (91.7%)	0.028
AngA dose (% target)	37.6 ± 31%	39.3 ± 28%	0.389
AngA > 25% target dose	37 (48.1%)	53 (48.6%)	0.94
MRA treatment	41 (53.2%)	69 (63.4%)	0.17
MRA dose (% target)	28.2 ± 30%	32.8 ± 29%	0.169
MRA > 25% target dose	30 (39%)	58 (53.2%)	0.055
AAD treatment	31 (40.2%)	50 (45.8%)	0.45
Amiodarone treatment	25 (32.5%)	27 (24.8%)	0.32
Digoxin treatment	15 (19.5%)	14 (12.8%)	0.3
BB+AngA+MRA treatment	26 (33.7%)	60 (55%)	0.0047
Follow up period (median (IQR), days)	1398 (567, 2361)	1400 (805, 2434)	0.27

**Table 5 jcm-10-01753-t005:** Cox proportional-hazards multivariate model for overall mortality.

Parameter	HR	95% CI	*p*
Age upon admission (years)	1.065	1.038–1.093	0.0001
Gender (male)	0.666	0.298–1.488	0.321
Renal disease	1.728	1.070–2.792	0.025
VA (any episode)	2.672	1.429–4.999	0.002
BB treatment	1.269	0.711–2.265	0.421
AngA treatment	0.515	0.285–0.929	0.028
MRA treatment	1.479	0.915–2.392	0.110

## Data Availability

The data presented in this study are available on request from the corresponding author. The data are not publicly available due to patient privacy.

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
