# Peer review of "Association of Guideline-Based Medical Therapy with Malignant Arrhythmias and Mortality among Heart Failure Patients Implanted with Cardioverter Defibrillator (ICD) or Cardiac Resynchronization-Defibrillator Device (CRTD)"

_jcm, 2021, doi:10.3390/jcm10081753_

Round 1

Reviewer 1 Report

 The research article, "Impact of Guideline-Based Medical Therapy on Malignant Arrhythmias and Mortality among Heart Failure Patients. 
implanted with a cardioverter defibrillator (ICD) or resynchronization device 
Resynchronization-Defibrillator Device (CRTD)" analyzed the association between the use of disease-modifying drugs and VA in patients with HFrEF. 

The paper is well designed and organized. 

My main concern is that the association between use and dose of disease modifying drugs is well known, and the paper does not add novelty to the existing literature.  

In addition, two important disease modifying drugs (ARNI and SGLT2-i) are not used in the study population making the treatment for heart failure analyzed in the study anachronistic

Author Response

Reviewer #1

The research article, "Impact of Guideline-Based Medical Therapy on Malignant Arrhythmias and Mortality among Heart Failure Patients implanted with a cardioverter defibrillator (ICD) or resynchronization device Resynchronization-Defibrillator Device (CRTD)" analyzed the association between the use of disease-modifying drugs and VA in patients with HFrEF.

The paper is well designed and organized.

My main concern is that the association between use and dose of disease modifying drugs is well known, and the paper does not add novelty to the existing literature.

In addition, two important disease modifying drugs (ARNI and SGLT2-i) are not used in the study population making the treatment for heart failure analyzed in the study anachronistic

Answer:

The reviewer explicates appreciation of our paper but expresses concern regarding its novelty. We thank the reviewer for bringing up the need to clarify the novelty of the paper in two major aspects: medication dose dependent effect on ventricular arrhythmias and the selected population of primary prevention ICD patients. Indeed multiple studies have shown the impact of GDMT on HF outcomes including HF symptoms, LVEF, LV remodeling, HF hospitalizations and total or cardiovascular death. Moreover, some studies have shown the dosage impact of these drugs on HF hospitalizations and death. Nevertheless, there is paucity of studies which examined the association of HF drugs, with emphasis on their dosage, to ventricular arrhythmias (VA) as a primary outcome. Moreover, only few studies examined the association of BB dosage in the specific subset of HF patients with primary prevention ICD/CRTD's. In response to the reviewer comment, we revised the introduction adding a sentence stating the need for further establishment of the direct relation of HF modifying medications with VA (p 3 bottom): "Nevertheless, the direct (non-circumstantial) impact of HF disease modifying medications and doses on VA as a primary outcome needs further establishment” (p3 bottom 1st paragraph). We also revised the discussion to emphasize the paucity of studies dealing with the association of BB dose with VA reduction, especially in subpopulation of primary prevention ICD/CRTD patients (p10 1st paragraph), having a brief summary of these studies. One of these was the Danish cohort study examining impact of BB dosage to VA in non-ischemic HF patients with primary prevention ICD/CRTD (ref 39), which was brought in our original discussion. Another study, recently published in Acta cardiol (ref 40), showed a significant association of up-titrating BB dose after primary prevention CRTD implantation with reduced appropriate device therapy. Lastly, a retrospective analysis of BB dosage among MADIT II ICD patients (ref 38) has shown significant association of higher BB dose with reduced appropriate ICD therapies. These studies were added to the revised manuscript (p 10 1st paragraph).

Notably, many HF patients implanted with primary prevention ICDs will never use their ICD and some may suffer from inappropriate shocks. Accordingly, examining the effect of HF drugs and their dosages on VA specifically is especially important as it may help clarify the additional advantage of ICD in presence of optimal HF treatment (which is still debated especially in non-ischemic CM). Thus, although our study is not the first to show BB dose association with reduced VA, it definitely adds robustness to the existing data.

The reviewer points that ARNI and SGLT2i were not used in the study population. ARNI was introduced into clinical practice in the 2016 ESC HF guidelines and was approved by the Israeli national medication committee at 01/2017, while SGLT2-i were only recently introduced as HF modifying drugs. Since study patients were enrolled during 2007-2017 they were not treated by these drugs. Indeed, this is still a limitation which is clearly noted in the paper limitations paragraph (p 12 1st paragraph) and in accordance with the reviewer comment we noted the fact that our study patients were not treated by SGLT2-i as well. Noteworthy, the same issue exists in some of the leading HF registries, for example in the CHAMP-HF registry published at 2018, where only 12.8% of eligible patients received ARNI. We agree that as new medications for HF gain acceptance the data accumulated on the effect of older more accepted treatment needs to be revised. However since in most countries and current registries the percentage of patients treated with ARNI and SGLT2i is still small, there is importance in accumulating the current experience with the available treatment at hand.

Reviewer 2 Report

Thank you for an original paper Impact of Guideline-Based Medical Therapy on Malignant Arrhythmias and Mortality among Heart Failure Patients Implanted with Cardioverter Defibrillator (ICD) or Cardiac Resynchronization-Defibrillator Device (CRTD) to be evaluated.

Please find my minor comments below:

  1. As you defined: medical treatment was determined based on medical prescriptions in the discharge letter of the index hospitalization – did you confirmed/analysed patients compliance in drug consumption?
  2. As the ICD/CRT-D programming was done according to team discretion your results are affected somehow but definitione of device intervention zones.  Did you assume to include in your analysis VAs that might occured in monitoring zones
  3. When multiple VAs occurred end-point was achieved with first VA detected
  4. Discussion is led very thoroughly and widely but an intersting aspect of indirect relations of BB dose ->VA->VA->mortality (though direct relationship between BB and mortality has not been achieved) should be emphasized in discussion perhaps

Author Response

Reviewer #2

Thank you for an original paper Impact of Guideline-Based Medical Therapy on Malignant Arrhythmias and Mortality among Heart Failure Patients Implanted with Cardioverter Defibrillator (ICD) or Cardiac Resynchronization-Defibrillator Device (CRTD) to be evaluated.

Please find my minor comments below:

1. As you defined: medical treatment was determined based on medical prescriptions in the discharge letter of the index hospitalization – did you confirmed/analysed patients compliance in drug consumption?

Answer: The reviewer raises a very important limitation of our study. We initially analyzed patients' compliance and changes to medication by evaluating patients' medication list on follow up visits (at least 6 months from the index hospitalization discharge). Nevertheless, as clinical visits timing varied widely compared with index hospitalization on one hand, and varied widely in relation to first event of VA on the other hand, we understood we should better stay with discharge medication list as this was the only homogenous timing, where we could evaluate all study patients in the same stage of their HF illness. Notably, in our initial analysis we found that 68% of patients did not change medication at follow up clinic visit (although even these patients had at times varying doses) while 32% had some medication added or withdrawn. The changing medications and the variable timing of these changes considerably complicate the ability to analyze consecutive events. To accommodate the reviewer point we emphasized this limitation in the revised manuscript limitation paragraph (p 12 1st paragraph) saying that: "Analysis of medication compliance and consistency could not be performed due to widely variable clinic visit timings, precluding homogenous time analysis of medications for all patients". Furthermore, we revised the methods section, clarifying that although follow-up visits were routinely scheduled every 6 months, in reality these visits varied widely in time (p5 middle of outcome paragraph).

2. As the ICD/CRT-D programming was done according to team discretion your results are affected somehow but definition of device intervention zones.  Did you assume to include in your analysis VAs that might occurred in monitoring zones?

Answer: Basically, our device programming was done in accordance with the expert consensus document accepted both by HRS and EHRA (ref 25), and we appreciate many centers are programming their patients' devices similarly. Nevertheless, this was mainly respected for the intervention zones while the monitoring zone was quite variable among patients. Thus, we included only VA detected via the intervention zones, which were constant among the study patients and enabled to study the impact of medication on VA occurrence. In our eyes, this fact was one of the strengths of the study, where only clinically-relevant 'serious' VA (necessitating device intervention) were evaluated. Nevertheless, to be clear, this was re-written both in the revised manuscript methods (p5 outcome paragraph) saying that: "During clinic visit, all VA episodes, necessitating device treatment (ATP, shock or both) were documented." and limitation paragraphs (p12 1st paragraph) saying: "only VA necessitating ICD/CRTD interventions were evaluated. Apparently, some VA had occurred in a monitor zone not necessitating any device intervention and thus were not included in our outcomes".

3. When multiple VAs occurred end-point was achieved with first VA detected

Answer: Indeed, for the Kaplan-Meier analysis, only first VA was relevant as this is a time-dependent analysis. For our univariate and multivariate analysis, all VA were relevant but as a rule we compared the group with one or more VA to the group with no VA.

4. Discussion is led very thoroughly and widely but an interesting aspect of indirect relations of BB dose ->VA->VA->mortality (though direct relationship between BB and mortality has not been achieved) should be emphasized in discussion perhaps.

Answer: The reviewer points out nicely that our results suggest an indirect relation of BB with mortality via effect on VA. Nevertheless, as there are plenty of large studies showing all-cause mortality reduction by BB, we felt this indirect relation does not add to the known literature in this aspect. Nevertheless, in the revised manuscript we did emphasize our contribution to the direct association between BB dose and VA, which was not evaluated extensively yet and the current data is less robust; and it is here were we find our manuscript to contribute most.

Reviewer 3 Report

Impression:

Hasin and colleagues prepared an interesting article that analyzes the association between GDMT utilization, with specific attention paid to GDMT dosage achieved, and clinical outcomes, mainly VA terminated by appropriate ICD/CRT-D therapy and all-cause death, among a population of HFrEF patients who received primary prevention ICD/CRT-D for the prevention of syncope and sudden cardiac death. The article reinforces the importance of uptitration of GDMT to clinical trial-level doses, not just the initiation of low dose GDMT, which is unfortunately far too common. The group finds that patients who received higher doses of appropriate multidrug GDMT experienced higher overall survival, and this seemed particularly true in patients who received RAAS inhibition and MRA >25% target dose.

Although the lessons in this study are important, there are certainly weaknesses that should be acknowledged.

  • This was a single-center retrospective study of 186 patients. Given the low overall population volume, it lends itself to type 1 errors (i.e. DM was associated with lower risk of VA and had trend towards higher overall survival).
  • The use of AAD in this primary prevention population was higher than seen in more contemporary populations and does not reflect routine current-day practice.
  • Most importantly, the authors do not explore the reasons that patients did not have uptitration of their GDMT agents. This is a very important point that can distinguish causation vs correlation. The findings as presented are correlative only; patients who have advanced HF often have difficulty tolerating higher doses of GDMT and are at higher risk of VA--- in this common scenario, the lower dose of GDMT did not CAUSE the VA, but rather is a marker of disease severity.
  • The authors should state early in the study, particularly in the Abstract and Methods, that this is a retrospective study. It is only first explicitly stated in the Limitations discussion at the end of the Conclusion.

Minor comments:

  1. In the Abstract, define ICD and CRT-D prior to using their abbreviations.
  2. In Line 43, “trails” should be changed to “trials”
  3. The text describes “Kaplan Myer” curves multiple times. This should be changed to “Kaplan Meier.”

Author Response

Reviewer #3

Impression:

Hasin and colleagues prepared an interesting article that analyzes the association between GDMT utilization, with specific attention paid to GDMT dosage achieved, and clinical outcomes, mainly VA terminated by appropriate ICD/CRT-D therapy and all-cause death, among a population of HFrEF patients who received primary prevention ICD/CRT-D for the prevention of syncope and sudden cardiac death. The article reinforces the importance of uptitration of GDMT to clinical trial-level doses, not just the initiation of low dose GDMT, which is unfortunately far too common. The group finds that patients who received higher doses of appropriate multidrug GDMT experienced higher overall survival, and this seemed particularly true in patients who received RAAS inhibition and MRA >25% target dose.

Although the lessons in this study are important, there are certainly weaknesses that should be acknowledged.

  • This was a single-center retrospective study of 186 patients. Given the low overall population volume, it lends itself to type 1 errors (i.e. DM was associated with lower risk of VA and had trend towards higher overall survival).

Answer: We thank the reviewer for his right comment and totally agree on this. Notably, in our original manuscript draft we dedicated a small paragraph discussing reduced VA in patients with DM. This was interesting as it seems counter-intuitive. Surprisingly though, DM patients with HF who were implanted with primary prevention devices were shown in previous studies to have increased SCD but with reduced appropriate shocks compared with non-DM HF patients, suggesting that the increased SCD incidence among HF patients with DM is due to non-shockable rhythms (example: Junttila MJ, Pelli A, Kentta TV, et al. Appropriate shocks and mortality in patients with versus without diabetes with prophylactic ICD. Diabetes Care 2020; 43: 196-200.). Nevertheless, considering our small population volume we decided not to include this controversial issue. According to the reviewer comment, we emphasized this issue in the revised limitation paragraph (p 11 bottom paragraph) including the following sentence: "single center data with low patient volume, exposed to type 1 statistical errors".

  • The use of AAD in this primary prevention population was higher than seen in more contemporary populations and does not reflect routine current-day practice.

Answer: The reviewer raises an interesting point. We appreciate most if not all of AAD's were given for treatment of atrial fibrillation (AF) which was quite frequent in our study (39%- Table 1). Notably, we excluded patients with prior sustained VA. Although the prevalence of AF in HFrEF population in general is usually lower, our prevalence was similar to other HF trials and registries (for example AF prevalence of 36% in the CHAMP-HF registry, and permanent AF prevalence of 24% in DANISH trial, suggesting higher percent of patients with permanent + paroxysmal AF). We attribute the high AF prevalence to our specific subpopulation of advanced HF patients in need of primary prevention device. Notably, Digoxin was quite frequent therapy, especially for chronic AF cases, in previous years. According to the reviewer comment, we noted this issue in the revised discussion (p 9, 1st paragraph) saying that: "Many patients (43.5% ) were treated by AAD, mostly Amiodarone and Digoxin, due to AF, which is prevalent in advanced HF patients (16,17).".

  • Most importantly, the authors do not explore the reasons that patients did not have uptitration of their GDMT agents. This is a very important point that can distinguish causation vs correlation. The findings as presented are correlative only; patients who have advanced HF often have difficulty tolerating higher doses of GDMT and are at higher risk of VA--- in this common scenario, the lower dose of GDMT did not CAUSE the VA, but rather is a marker of disease severity.

Answer: We thank the reviewer for this crucial comment. Indeed, as this was a retrospective study going over patients' medical records, it was not always clear why the dosages were low in each specific patient. Due to its importance, we dedicated the Discussion's 5th paragraph to this issue. As we write in this paragraph (p 11 upper paragraph), although we do not have the explicit reason for each patient's low dose (sentence added to the revised paragraph), we suggest this was due to suboptimal medical surveillance, since most patients in our study had tendency for relatively high blood pressures with heart rates between 70-80 bpm. Thus, at least regarding BB medication, we could suggest that patients' BP and heart rates were not 'obstacles' for treatment. Furthermore, we found that only minority of study patients was followed in HF clinic, where there were significantly more patients treated by HF medications and specifically with significantly increased doses of BB (as written in the Results, p6 bottom paragraph). Again, suggesting that the low dose of most study patients was not due to their illness but rather non optimal treatment. This issue of increased prevalence and dosage of BB in patients followed at HF clinic was mentioned in the Results but not clearly mentioned in the original discussion. Accordingly, we mention it in the revised discussion (p11 upper paragraph). On the whole, we agree that the above is circumstantial evidence only and this was explicitly written in the revised discussion (p11 upper paragraph) with the summary sentence of: "Importantly, as we do not have the clear reason for low medication dosage in each patient, we could not rule out low dosage secondary to advanced HF stage. Nevertheless, we do suggest from the above circumstantial data, that this was not the case".

Due to the above and in line with the reviewer comment we made several changes in the revised manuscript to clarify we suggest a significant association between BB dosage and VA, and NOT of a cause and effect relationship. These changes include:

  1. Title changed from "impact of…" to :"Association of …" as we acknowledge "impact" may suggest a cause and effect relation.
  2. The abstract (p2) and introduction (p3 bottom paragraph) were changed similarly, where we replaced "impact on" by "association with"
  3. A clear sentence was added to Discussion (p 11 upper paragraph) stating that we could not rule out low medication dosage secondary to advanced HF, although our circumstantial data suggest the opposite.
  • The authors should state early in the study, particularly in the Abstract and Methods, that this is a retrospective study. It is only first explicitly stated in the Limitations discussion at the end of the Conclusion.

Answer: We thank the reviewer for this important comment. Accordingly, this was explicitly written in the revised abstract (Abstract methods, p2) and methods section (p 4) with the following sentence: "Retrospective single center analysis of HF patients implanted with primary prevention ICD/CRTD to evaluate the association between HF modifying drugs, with emphasis on their doses, to VA and total mortality".

Minor comments:

  1. In the Abstract, define ICD and CRT-D prior to using their abbreviations.

Answer: According to the reviewer comment, these were defined in the revised abstract.

  1. In Line 43, “trails” should be changed to “trials”

Answer: Corrected.

  1. The text describes “Kaplan Myer” curves multiple times. This should be changed to “Kaplan Meier.”

Answer: Corrected.

Round 2

Reviewer 1 Report

The reviewer thanks the authors for their response

The reviewer remians of the opinion that the topic of the study is not very  original (for example the use of beta-blockers uptitration on VA and ICD shock in patients with ICD/CRT-D was investigated by Masarone et al. Med Sci (Basel). 2019 Jun 18;7(6):71).

However the effort of the authors to adress the reviewer concerns are appreciable.

Reviewer 3 Report

No further comments. I commend the authors on their work in preparing this manuscript.